# A Tale of Two Audiences: Formative Research and Campaign Development for Two Different Latino Audiences, to Improve COVID-19 Prevention Behavior

**DOI:** 10.3390/healthcare11131819

**Published:** 2023-06-21

**Authors:** Dianna Bonilla Altera, Imani Cabassa, Genevieve Martinez-Garcia

**Affiliations:** ICF, Reston, VA 20190, USA; dianna.bonillaaltera@icfnext.com (D.B.A.);

**Keywords:** health communications, formative research, COVID-19, vaccines, Latino/Hispanic

## Abstract

The COVID-19 pandemic disproportionately affected the Latino population in the United States, further exacerbating the existing racial and ethnic health disparities that this group faces. While government health entities rushed to develop COVID-19 prevention educational materials in Spanish, these failed to recognize the unique motivators and barriers that move different Latino audience segments to act. We conducted five online focus groups with two different Latino audience segments, general Latino people, and Latino migrant workers, to assess their experience navigating the pandemic, their engagement in preventive behavior, and their consumption of health news. While the general Latino audience had higher levels of social capital and established preventive healthcare, they were more skeptical about getting the COVID-19 vaccine. Migrant workers needed to be vaccinated to retain their jobs, and saw the vaccine as the only way to keep their families healthy. We used the focus group results to develop two different creative concepts that aligned with each audience’s unique experience. Our study highlights the importance of developing hyper-focused messages, responsive to the experience of distinct audience segments, for maximum impact.

## 1. Introduction

The COVID-19 pandemic disproportionately affected the Hispanic/Latino (hereafter Latino) population, further exacerbating the existing racial and ethnic health disparities that this group faces. Latino populations were 1.5 times as likely as White populations to contract COVID-19, shared a greater proportion of COVID-related hospitalizations, and made up 20% of COVID-related deaths [1,2]. Additionally, Latino migrant workers have also suffered the burdens of the pandemic. Specifically, migrant workers make up a disproportionate share of meat-processing workers. A study investigating the five largest meat-processing companies in the United States revealed 59,000 COVID-19 cases and 269 deaths linked to the various plants in the pandemic’s first year alone [3]. In some plants, nearly half, or more, of all employees had contracted COVID-19 by February 2021 [3]. Scholars, researchers, and healthcare professionals have cited multiple reasons for the disproportionate burden of risk, illness, and death among Latino people. These include various social determinants of health, such as education, housing, and access to healthcare [4]. Latino people were overrepresented as essential workers during the pandemic, which likely led to increased exposure to the virus [5,6]. However, another factor that created health inequities related to COVID-19 for the Latino community was the widespread misinformation about the virus [7].

At the onset of the COVID-19 pandemic, the U.S. government struggled to communicate about the pandemic. When dealing with the novel virus, initially there was a lot of uncertainty, fear, and anxiety, coupled with information that was constantly changing, based on protecting the masses [8]. The World Health Organization declared COVID-19 a pandemic on 11 March 2020, and soon after, the United States began implementing a number of control measures, including quarantining practices, social distancing, and cancelling large gatherings that put people at risk [8]. Implementing these measures proved to be difficult, because as public health scientists learned more, measures constantly changed, or misinformation was spread. While trying to communicate safety measures, the United States and other nations also faced an infodemic, where misinformation or disinformation was being spread globally through social media platforms [8]. Toward the end of 2020, pharmaceutical companies had developed COVID-19 vaccines and, with emergency authorization from the Food and Drug Administration (FDA), were ready to administer these vaccinations to the public [9]. However, this brought about many more communication challenges, including conspiracy theories, misinformation, and public fear of the quick development of the vaccines [10].

Some scholars have begun studying the role of public health messaging and communication tactics during the onset of the COVID-19 pandemic. The news media originally played a role in delivering valuable life-saving information; however, as time went on, inconsistent messaging from the media and public leaders led to the public feeling confused about how to handle the pandemic [11]. Along with the failure to deliver consistent information, another issue was making sure that the public health messaging materials were accessible to the Latino community. For many Latino people, this was not the case with COVID-19 communication materials. A qualitative study that looked at the experiences of Latino patients who were hospitalized because of the virus described how health communication materials impacted their risk of the virus [12]. When describing their experiences, many Latino people mentioned receiving their COVID-19 information from social media, but also feeling as if the virus were made up by the government to identify undocumented individuals [12]. Others discussed how there was a lack of information, leading them to feel confused by the different recommendations they were given, or as if the virus were made up [12]. During the onset of the pandemic, communication materials also lacked cultural and linguistic competency and sensitivity. One study discussed how, initially, communication materials focused on individual behavior choices, making COVID-19 prevention and protection part of an individual’s responsibility [13]. However, when it comes to Latino communities, individuality is not the norm; instead, collectivity and togetherness are celebrated. In COVID-19 communication messaging, collective messaging was often valued, yet hard to find [13].

Health information and materials should be accessible, attainable, and understandable to the audiences they are intended for, in order to be useful [14]. This also means that different messages and materials could be needed for different subgroups within a large audience. While it was necessary to create messages and materials that resonated with the Latino general audience, Latino migrant workers—a subgroup of this community—also needed messages and materials developed for them. Tailoring health messaging for different subgroups of an audience allows the message to fit personal needs, and is often more effective at influencing health behaviors [15]. Within the Latino community, while some values remain the same across the entire audience, subgroups (such as Latino migrant workers) often have different traits and nuances that allow them to be motivated by different messages [16]. Because Latino migrant workers were considered essential workers during the COVID-19 pandemic, and therefore at an increased risk of being infected with the virus, it was important that messages be developed specifically for this subgroup of the Latino community, while also developing COVID-19 messages for the Latino community as a whole. Faced with the lack of tailored communication to effectively reach different Latino audiences, our study aimed to understand the COVID-19 communication needs of two Latino audiences, and develop materials tailored to them.

### Study Goals

This study is a multiphase, national study to develop and test audience-informed COVID-19 prevention messages for two different Latino audience segments—a Latino general audience, and Latino migrant workers. This article will discuss the findings from the Phase 1 formative study, and the subsequent concept development (Phase 2). The goals of Phase 1, a qualitative formative study, were to learn about the perceptions, knowledge, and attitudes toward COVID-19 vaccination, and to identify priority communication channels, for the two Latino audiences. These data were then used to guide the development of audience-specific prevention messages, and two creative concepts per audience (Phase 2). During Phase 3, we tested the creative concepts with audience members through eye-tracking testing and interviews, to identify the concept that resonated best. (Phase 3).

Our formative research with Latino audiences is part of a larger research-based communications project to disseminate audience-informed COVID-19 prevention materials to 11 different audiences who bear a disproportionate burden of COVID-19 infections. These include different racial and ethnic groups, Native communities across the United States, migrant groups, and individuals with disabilities. The project was led by [NAME OF SCHOOL OF MEDICINE], and the full team included multicultural communication partners from academic institutions, community-based organizations, private firms, and subject matter experts working with the priority audiences. The project engaged local communities to provide guidance, throughout the research and creative development.

## 2. Materials and Methods

Between February and April 2021, we conducted five online focus groups with thirty Latino adults from two audiences: the general Latino community (two groups, n = 18), and migrant workers (three groups, n = 12). At the time of the focus groups, COVID-19 vaccines were not yet available to the general population. Our study protocol was approved by the Morehouse School of Medicine Institutional Review Board (IRB Protocol 1645842-2). We specifically recruited Latino individuals who fell into two categories, “general Latino”, and “migrant workers”. To be considered “general Latino”, participants had to self-identify as Latino regardless of place of birth, be at least 18 years old, and not work in communications, marketing, or any health field. To be considered a “migrant worker” specifically, self-identified Latino individuals also had to work in either farming, agriculture, meatpacking, or the dairy industry, and not in managerial positions within these industries. For the purpose of this study, we are referring to each audience as “general Latino” and “migrant worker”.

### 2.1. Recruitment

We employed multiple strategies to recruit participants from the two audiences. To recruit Latino participants from the general Latino population, we used a research firm that recruited participants from a preexisting audience panel. The firm sent email invitations with a screener link to eligible panel members. They enrolled and scheduled individuals who met all eligibility criteria. To recruit Latino migrant workers, we engaged an academic partner that provided direct services to migrant workers. We used a purposive sampling approach, posting flyers in areas frequented by migrant workers, and using word of mouth and referrals. Any staff engaged in the recruitment, enrollment, and scheduling of focus groups were certified in human-subject research. In addition, recruiters completed a one-hour training session to learn about the recruitment process, including identifying and screening participants, and scheduling each participant for a focus group.

### 2.2. Implementation

We conducted all focus groups online, to allow participants to dial or video call in using their cellphones or computers. A trained facilitator, who self-identified as Latino and was a native Spanish speaker, conducted the 90-min groups. Although we offered groups in English and Spanish, all groups were conducted in Spanish, as preferred by the participants. We recorded and transcribed the focus groups verbatim, and deidentified the data prior to analysis.

### 2.3. Theoretical Framework

The aim of the study was to learn about the motivators and barriers to COVID-19 prevention, and to identify the best way of disseminating prevention messages, to these communities. We developed a moderator’s guide to address three main topics: COVID-19 testing, COVID-19 vaccination, and information sources. We used the Health Belief Model (HBM) as our core theoretical framework to guide the development of the focus group and our analytical approach [17]. HBM (Table 1) is a value expectancy theory with six components, that proposes that people will take action to prevent a disease if they (1) believe themselves to be susceptible to it (perceived susceptibility), (2) believe it will result in serious consequences (perceived severity), (3) believe that a preventive action is available to them (cues to action), (4) feel confident in taking action (self-efficacy), (5) believe that the barriers or cost of actions are outweighed by the benefits (perceived barriers and perceived benefits), and (6) believe that the adoption of the behavior will result in positive outcomes (perceived benefits). This theory has been used consistently to study vaccination hesitancy in general, and COVID-19 vaccination specifically [18,19]. In addition, we collected information on preferred health information sources and channels, to help inform the development of the creative concept and its dissemination.

### 2.4. Data Analysis

We used NVivo 12 to conduct a thematic analysis of data, using a codebook with categories identified *a priori* based on the Health Belief Model, and subsequently coded by emerging themes. Two coders independently coded the data, and met to organize emerging themes and reconcile differences.

## 3. Results

### 3.1. Phase 1 Results

We recruited 30 adult participants (15 males and 15 females) who met the eligibility criteria of our two audience groups (general Latino n = 18; migrant worker n = 12) (See Table 2). Despite our best efforts to recruit an equal number of participants across our two samples, it was more challenging recruiting migrant Latinos, resulting in a smaller sample. Most individuals in our sample were born outside of the United States, in a Latin American country, and had a strong preference for Spanish. However, there were notable differences between the two Latino audiences, in terms of social capital, insurance level, and the impact of their experience with COVID-19. General Latino participants had migrated from different countries in the region (e.g., Argentina, Colombia, and Cuba) and almost half had at least some college or trade school education, and reported preferring English equally or less than Spanish (44%). Most had health insurance (94%), and had gotten the flu vaccine in 2020 (72%). Most migrant workers, on the other hand, were born almost exclusively in Mexico, had a high school degree or less (83%), and did not have insurance (58%). Most migrant workers reported not receiving the flu vaccine in 2020 (75%). The two audiences reported a different experience with COVID-19. While the general Latino audience reported not having a close experience with COVID-19, all migrant workers reported having experienced COVID-19 either personally, or through a peer or family member.

### 3.2. Behavioral Motivators

#### 3.2.1. Perceived Susceptibility

Perceived susceptibility varied sharply among the two groups. General Latino participants reported being in good health and having low exposure to COVID-19 cases, thus they reported a low perceived susceptibility to the virus. Participants reflected on the exposure risk from their family and close acquaintances. As one participant from the general Latino group highlighted, “We are five in the family, three kids, my wife and I [and] people close to us haven’t had COVID. If anyone we are frequently around find out they are positive, then [the vaccine] would be worth it (General Latino, Male)”. On the other hand, migrant workers reported being exposed more often to COVID-19, and experiencing a high number of COVID-19 cases and deaths among their community members; thus, they reported a high perceived susceptibility to the virus. In the words of one migrant worker, “Well, [I] mainly got vaccinated because I was positive for COVID before and lost three relatives to it this year, I felt responsible for those around me, especially my children (Migrant Worker, Female)”. See Table 3 for a summary of results by theoretical construct.

#### 3.2.2. Perceived Severity

The severity of COVID-19 was perceived highly among migrant workers, who reported feeling concerned about their own health and the health of their family and friends. They commented on the emotional impact of witnessing the death of loved ones due to COVID-19. As one migrant worker described, “[COVID] affected me in the sense that my daughters could no longer visit me. My granddaughters would visit me almost every day. After that, my granddaughters couldn’t visit me as they used to. They’d stay outside and stand at the door, they couldn’t come in. They couldn’t give me a hug or anything (Migrant Worker, Female)”.

Conversely, general Latino participants commented more often on the social impact of COVID-19. They reported, for example, the importance of developing and strengthening interpersonal relationships, the negative impact of missing school on children, and the impact it had on older Latino adults who care for young children. One Latino participant commented, “My bigger concern is for my kids that are at school age and it’s really frustrating to see how they are used to go to school and being surrounded by their friends. Now they are in front of a machine where there’s no personalized attention from the teachers. It’s a stress at the house, it frustrates them, it frustrates us. (General Latino, Female)”.

#### 3.2.3. Perceived Barriers/Self-Efficacy

Participants faced different barriers to receiving COVID-19 vaccines and accessing testing sites. The most common barriers that were reported included access to testing sites, vaccine hesitancy due to fear of the long-term effects of COVID-19 vaccines, and their immigration status.

Accessing testing and vaccination sites: both audiences reported high self-efficacy and low perceived barriers related to locating and accessing COVID-19 testing sites. One general Latino participant explained their process, “For my test, all I did was put my ZIP code. I looked for places around my area, it showed you different parking lots or parks and then you could schedule the appointment right in one. I found it to be very easy because I just added my information and my telephone number, my email, they sent me a confirmation and that was it (General Latino, Female)”. However, their families and friends, especially older adults, had limited access to these testing sites, due to a lack of transportation or of access through employer-based testing sites. Additionally, they reported that their family and friends had low proficiency with technology, and faced challenges in using an online platform to schedule tests. Migrant workers mentioned that they had access to employer-based testing sites. Although locating and scheduling appointments for COVID-19 was not perceived as a barrier for them personally, they reported that the lack of materials in Spanish presented a challenge. As one migrant worker explained, “Many elderly people work in my area and getting tested was hard for them because they couldn’t get an appointment. To get registered for the vaccines, most of it involved doing it online…but you needed to have an email address or something like that. Many people don’t have access to or don’t understand the internet. It was hard for them (Migrant Worker, Female)”.

Experiencing vaccine hesitancy: participants across both audiences expressed hesitancy to get vaccinated, due to fear of the long-term effects of the vaccines, misinformation and myths surrounding COVID-19 and the vaccines, and distrust in science. However, their perspectives and responses varied among the two audiences. General Latino participants most often mentioned unknown long-term effects, vaccine misinformation, and conspiracy theories as reasons not to receive a COVID-19 vaccine. As one participant explained, “I don’t just want to do something I have no idea about. Because I know that [the vaccine] has side effects and I have heard in the news of many things that have already happened and many things that are not being disclosed (General Latino, Male)”.

General Latino participants often commented that they would use a “wait and see approach” before being vaccinated. As one participant expressed, “I’d like to wait a bit more for later in the future. But I’ve decided not to get it because there are a lot of unknowns still of what could happen or what the vaccine could cause in the future (General Latino, Female)”.

Migrant workers were also concerned about the vaccine’s safety, but trusted that receiving the vaccine would do more to protect them and their families than to cause harm. As one migrant worker reported, “I was very scared because I heard many people had complications and died due to the vaccine… I heard a lot of bad things, but also good things, and I have a child that has asthma… If we don’t get vaccinated, it would hit us stronger (Migrant Worker, Female)”.

Reporting on immigration status: the barrier that was mostly mentioned by migrant workers was the fear of being asked their immigration status to receive a COVID-19 vaccine. Participants from the general Latino group did not report fears of providing immigration information as a barrier to being vaccinated. As one migrant worker recalled, “I have heard that some people were asked for [an ID]. However, when I went, I was scared. I was like, ‘What if they ask me for something?’ Thank God, they didn’t ask for anything. All they asked for was a home address and some personal data (Migrant Worker, Female)”.

#### 3.2.4. Perceived Benefits

Participants in both groups had different perceptions of the benefits of being vaccinated. Most participants in the general Latino group did not report any benefits stemming from the COVID-19 vaccines. The few participants that mentioned a benefit reported benefits to society as a whole, such as stopping the pandemic and protecting family members. As one participant explained, “I mainly believe in vaccines because they are the ones that are going to help us solve this problem. I’m waiting for my turn to come so I can do it. As soon as it’s my turn, I will do it because it’s the only way to protect us (General Latino, Male)”. Migrant workers, on the other hand, reported that the vaccines protected their family and their community. Despite the barriers this group faced, they perceived that the benefits of the vaccines outweighed the obstacles. As one migrant worker put it, “My daughter took me to get vaccinated because she worries about me. I did it to be responsible, for my benefit and the family’s benefit in general (Migrant Worker, Female)”.

#### 3.2.5. Cues to Action

At the time of this study, the vaccine was not yet available to all adults, only to frontline workers, such as the migrant workers. Migrant workers got vaccinated due to employment requirements. However, they reported more often that they were vaccinated because they had experienced symptoms, and because they had witnessed firsthand the impact of COVID-19 on members of their community, and felt responsible for protecting children and older adults, and stopping the spread of COVID-19. As one migrant worker described, “Well, I also got vaccinated so that my family wouldn’t get infected again, considering my father-in-law and my daughter had a hard time with it. I didn’t want to infect them (Migrant Worker, Male)”.

The general Latino participants showed less interest in getting vaccinated. Participants also reported less exposure to COVID-19, and less perceived need to getting tested. Those who got tested most often reported getting tested due to employment requirements, and to have peace of mind when visiting elderly relatives. As one participant reported, “All of a sudden, they closed the office and said that there was a case. Then we all had to go get the test and [they told us] that we couldn’t go back until we had a negative test result (General Latino, Female)”.

#### 3.2.6. Communication Preferences

General Latino and migrant worker participants reported different communication preferences. The Latino general audience reported relying more on English and Spanish national and local television news channels (e.g., CNN, Univisión, Telemundo, ABC News, Apple News, BBC Network, or KTLA Network in Los Angeles), social media (e.g., Facebook, Instagram, YouTube, Snapchat, TikTok, and Reddit), and family and friends via in-person conversations or WhatsApp. A few participants mentioned Spanish radio stations and internet searches.

On the other hand, the migrant workers consistently relied on local Spanish news outlets (e.g., Univisión and Telemundo), outreach by *promotores de salud* (community health workers), and local clinics to stay informed about COVID-19. Migrant workers mentioned less often engaging in social media and internet searches, due to their limited access to reliable internet service. One major distinction in the dissemination channels between the two audiences was the migrant workers’ reported reliance on community organizations and businesses for COVID-19 information. Migrant workers mentioned community health centers, local supermarkets, and churches as dissemination channels, where they saw information about COVID-19 posted that helped them to stay informed. There was no one physical space that was highlighted more predominantly than others. However, the general Latino participants did not mention any community organization as a trusted information source.

The type of materials the audiences preferred to receive about COVID-19 testing and vaccination varied slightly. Migrant workers reported that they preferred receiving printed materials, in-person-delivered to their home, or posted in community centers they frequently visit and trust, or via phone call or text message. The general Latino participants preferred materials sent via mail, or messages on billboards, public transportation, and radio.

#### 3.2.7. Phase 2 Concept Development

We developed two concepts, one for each audience, that focused on the main motivators to engage in COVID-19 prevention behaviors. While the general Latino concept focused on the importance of protecting community elders, the concept for migrant workers focused on their role of sustaining their families. To develop these evidence-based concepts, we used multiple data sources to paint a complete, nuanced picture of each audience. We combined focus group data with an audience-specific profile that included sociodemographic and cultural attributes of each audience. These profiles used data from the U.S. Census Bureau; peer-reviewed and grey literature; government, business, and organization reports; and marketing reports on various disciplines, such as economics, education, employment, acculturation, health literacy, influencers, media/social media consumptions habits, spendings habits, and entertainment. Once the audience profile was completed for each audience, we identified one main message, and four supporting messages, to underpin the creation of the concepts. Our project partner, a university located in the southeast United States, tested each concept with audience members, using eye-tracking and facial-expression monitoring, and open-ended reflection questions, to assess concept appeal, message retention, intention to act on the call to action, and message complexity. We developed a complete suite of communication materials for the winning concepts, taking into account the preferred communication channels for each audience, and the dissemination capacity of our community partners. Table 4 lists the materials created for each audience.

### 3.3. Creative Materials for the General Latino Community

The final creative package developed for the general Latino audience is titled “They’re Waiting for You”. This concept focused on the importance of grandparents in Latino families. The focus group data revealed important motivators and barriers to getting vaccinated. Despite exhibiting vaccine hesitancy due to mistrust in science and fear of vaccine side effects, this audience was motivated by their desire to spend time with family members and elders. Based on this information, we developed one primary message, and four creative concepts, that encouraged COVID-19 vaccination as a pathway to spending time with family, especially elders. Rather than focusing on the risks of COVID-19, the message emphasized the positive outcome of vaccination: Getting vaccinated gets us back to life and to our family and friends, whom we’ve missed so much over the last year. The creative concept used photography to accentuate the primary message that vaccines would get us back to the people who needed us, such as our grandmothers. It used an emotional motivator to acknowledge that grandparents were waiting for their family members to be a part of their lives again and that, to do that, the community needed to be vaccinated.

### 3.4. Creative Materials for the Migrant Worker Community

The final creative package developed for migrant workers is titled “You’re Their Rock” (See Figure 1). This concept focused on the critical role that migrant workers play in sustaining their families, in the United States and abroad. To develop this concept, we relied on the focus group data, and the economic and employment data from the audience profile. A recurrent theme from the focus group discussion was the audience awareness of their exposure to COVID-19 through neighbors, co-workers, and family members. Despite also feeling mistrust toward science, and fear of any long-term negative effects from the vaccine, migrant workers had a strong desire to protect their family from illness and potential death. Access to testing sites, and vaccination through their employment, made it easier for them to get vaccinated, and helped them to overcome the barriers of scheduling appointments online, and fears of showing proof of residency. However, from the audience profile, we know that migrant workers also help support family members on both sides of the border. Therefore, this concept served as a reminder that the people they loved the most were relying on them. At the time of this study, migrant workers were eligible for vaccination, as they were considered essential workers. All migrant worker participants had received at least one dose of a COVID-19 vaccine. However, it was important to emphasize the need to be fully vaccinated, which entailed up to two doses, depending on the vaccine.

Based on these data, the overarching key message for the concepts centered around the importance and dependence of family: *Your family needs you. Get fully vaccinated (See Figure 2).* The message emphasized their critical role as provider to their family. This primary message used an emotional motivator, with a straightforward message: *Your family needs you.*

## 4. Discussion

This study gathered valuable insights on COVID-19 prevention behaviors from two segments of the Latino audience in the United States, with the aim of developing audience-centric communication materials. To our knowledge, this is the first study that attempted to dive deeper into the motivators and barriers that Latino audiences faced to prevent COVID-19, and that used these insights to develop an audience-centric campaign. Moreover, our study was conducted between February and April 2021. During this period, frontline workers, such as healthcare professionals and workers in the agro-industry, were eligible to get vaccinated. This period was a critical time that allowed us to study audiences’ perceptions of COVID-19 vaccination before it became widely available.

Our study revealed commonalities among the two Latino audiences, and important differentiators that drove their COVID-19 protection behaviors. First and foremost, both were hesitant to get vaccinated due to mistrust toward science and the government, and fear of the long-term effects of the COVID-19 vaccine. Other studies conducted around the same time have consistently shown how Latino people exhibit higher levels of vaccine hesitance, and lower likelihood of getting vaccinated, than other groups [20,21]. However, in our study, participants focused more on their lack of trust in the vaccine development process, and on the presumed deaths resulting from the vaccine. Misinformation about the vaccine fueled their mistrust. In fact, misinformation about COVID-19 was widespread through social media at the time of this study [8,12]. However, participants’ resistance to the vaccine was tempered by strong motivators. While for the migrant workers, getting vaccinated was considered critical to protecting and providing for their families, for the general Latino audience, the vaccine was a pathway to going back to their normal family life.

Our study showed that this reaction was greatly due to how each audience experienced COVID-19. Migrant workers reported high exposure to COVID-19 due to proximity to their co-workers, family, and community members. In fact, all the migrant worker participants reported knowing people who had either been sick or had died from COVID-19. Moreover, migrant workers were eligible to get a vaccine due to their status as essential workers, had access to testing and vaccination through their employment, and feared not being able to stay employed or find employment if they were not vaccinated. Our finding was consistent with other studies showing that having low income, and knowing someone with COVID-19, increases vaccine acceptance [22]. Having an enhanced sense of susceptibility, by knowing someone who has been sick or died from COVID-19, in addition to employment requirements, were motivators that led this group to get vaccinated despite their hesitancy. The general Latino audience, on the other hand, perceived a low risk to contracting COVID-19, and reported not knowing anyone in their social network who had become ill or died. Moreover, many participants from this group reported that they worked from home and had the ability to maintain social distance. At the time of the study, none of the general Latino participants were vaccinated, as vaccines had recently become available to the general population. The scheduling of vaccines was carried out through online platforms, and it was challenging to find appointments. Despite their mistrust and fear of the vaccine, the general Latino audience was motivated by their desire to spend time with family members, particularly their elders.

### Limitations

Our study was qualitative in nature, and gathered experiences from 30 individuals. Thus, our results cannot be generalizable to the entire Latino population in the United States. Our discussions coincided with the release of the COVID-19 vaccines for specific groups. The discussion with the general Latino group took place in February 2021, while the discussion with the migrant Latino group took place in April 2021. Due to the rapid changes in vaccine availability, it is possible that the differences noted between the groups were due to the vaccine availability, or new information about the vaccines coming to light, between February and April 2021. The different recruiting mechanisms may have introduced biases in our key sample demographics. The participants in the general Latino group were recruited from across the United States, and were part of an existing panel of research participants. On the other hand, the migrant workers were recruited by one academic partner located in Texas, and were recruited through flyers and word of mouth. The fact that the general Latino participants were part of a research panel may have contributed to them having relatively higher social capital, and skewing slightly younger than the migrant workers. Migrant workers were considered essential workers, and received vaccination information and testing more often. Moreover, the vaccine was available to them sooner. Early access to the vaccine, and easy access through employment, may have influenced their perceptions about vaccination.

Despite these limitations, our study was successful in achieving its main aim, which was to gather critical insights to develop audience-focused campaigns. Despite our relatively small sample size, and some of the biases that may have been introduced, as described above, our findings mirror the findings from other studies conducted during a similar period. We acknowledge that focus group data can only take us so far in terms of understanding the audience. For this study, we complemented our focus group findings with secondary data, to create a full audience profile that combines sociodemographic and psychographic information, to fully understand the audience’s social and cultural context, and make sense of the research findings. Scholarly studies that aim to gain a deeper understanding of the audiences, and conduct robust theory-based analyses, should consider engaging a larger sample of participants.

## 5. Conclusions

Our study shows how the generalized information about health-related emergencies (e.g., COVID-19) represents a missed opportunity to connect and empower audiences such as the Latino community to take preventative measures, due to lack of cultural relevancy, lack of materials in Spanish, and lack of messaging relevance that resonates directly with the community [13]. Our study demonstrated that broadly defined audiences are not homogenous in their motivations, barriers, and perceptions related to health behaviors. While both Latino audiences prioritized their family as a key motivator, each Latino audience segment experienced the pandemic differently, which influenced their ability and intention to engage in COVID-19 prevention behaviors. Communication materials aimed at changing behavior must account for these differences, in the creative process, to increase their reach and impact among specific audience segments.

## Figures and Tables

**Figure 1 healthcare-11-01819-f001:**
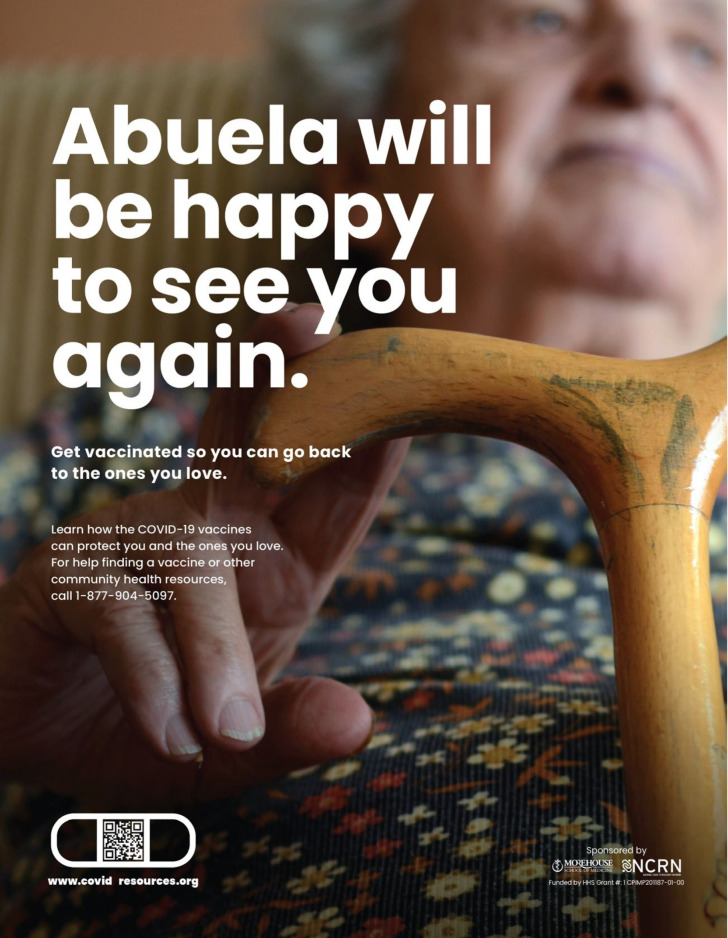
General Latino Concept.

**Figure 2 healthcare-11-01819-f002:**
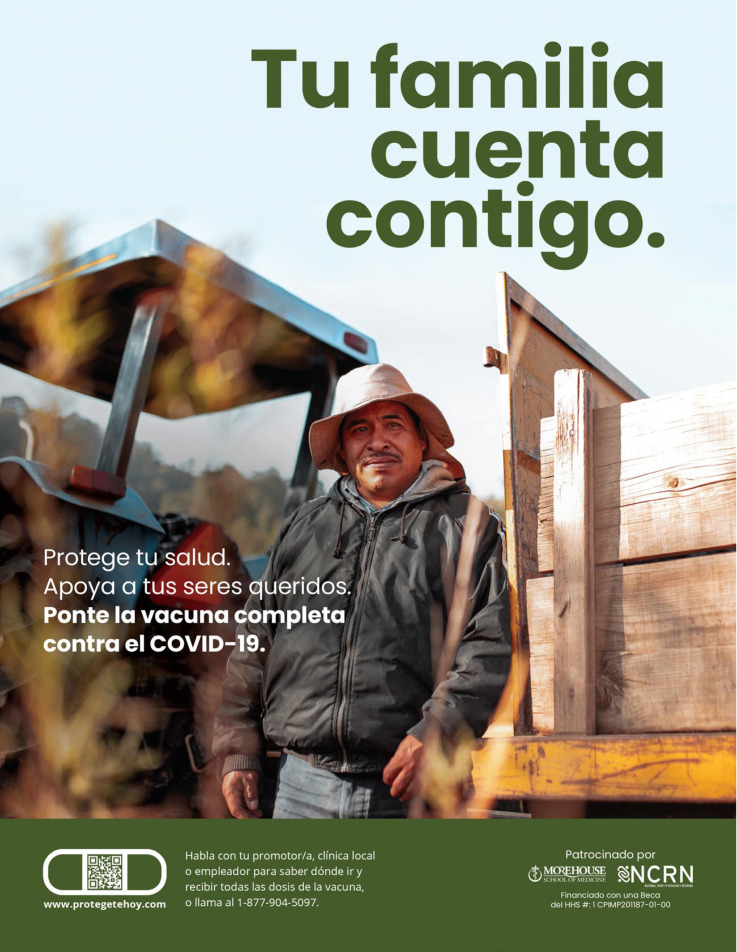
Migrant Worker Concept.

**Table 1 healthcare-11-01819-t001:** Health Belief Model operationalization of constructs.

Theoretical Construct	Operationalization	Question Example
Perceived susceptibility	Belief about the likelihood of contracting COVID-19.	Those who have not gotten a COVID-19 test, for what reasons have you not gotten tested?
Perceived severity	Beliefs about the negative consequences of contracting COVID-19.	In what ways, if any, does the COVID-19 pandemic affect you?
Perceived barriers	Perceived obstacles to receiving a COVID-19 test or vaccine.	What, if anything, has prevented you from getting tested?
Perceived benefits	Belief about the positive outcomes associated with receiving a COVID-19 vaccine.	How might you benefit by receiving a COVID-19 vaccine?
Cues to action	Stimuli needed to trigger the decision-making process to get a COVID-19 test or vaccine. These cues can be internal (e.g., symptoms of COVID-19) or external (e.g., policies, employment requirements).	For those who have gotten the COVID-19 test, tell us what prompted you to get tested?
Self-efficacy	Confidence in one’s ability to perform the recommended behaviors.	How likely are you to get the COVID-19 vaccine in the future?
Information sources	Most frequented and trusted sources to obtain COVID-19 information.	If you were looking for information on COVID-19 testing or vaccination, where would you look?
Preferred channels	Communication formats, channels, and venues where COVID-19 information is sought.	How do you usually receive COVID-19 testing or vaccine information?

**Table 2 healthcare-11-01819-t002:** Demographic characteristics.

Characteristic	Latino—General		Latino—Migrant Worker	
	#	%	#	%
Total	18	100%	12	100%
Gender identity				
Male	9	50%	6	50%
Female	9	50%	6	50%
Age				
18–24 years	2	11%	-	-
25–34 years	4	22%	1	8%
35–44 years	6	33%	3	25%
45–54 years	4	22%	4	34%
55–64 years	2	11%	3	25%
65 years and older	-	-	1	8%
State of residence				
California	9	50%	-	-
Florida	9	50%	-	-
Texas	-	-	12	100%
Country of birth				
Argentina	1	6%	-	-
Colombia	1	6%	-	-
Cuba	3	17%	-	-
Guatemala	1	6%	-	-
Mexico	8	44%	11	92%
Nicaragua	2	11%	-	-
United States	-	-	1	8%
Venezuela	2	11%	-	-
Preferred language				
Only Spanish	6	33%	12	100%
Spanish more than English	8	44%	-	-
Both equally	4	22%	-	-
English more than Spanish	-	-	-	-
Only English	-	-	-	-
Education level				
Less than high school degree	3	17%	7	58%
High school degree, GED, or other credential	4	22%	3	25%
Some college or trade school but no degree	3	17%	2	16%
Associate’s or trade school degree	4	22%	-	-
Bachelor’s degree	3	17%	-	-
More than a bachelor’s degree	1	6%	-	-
Health insurance status				
Have	17	94%	5	42%
Do not have	1	6%	7	58%
Close COVID-19 contact *				
Yes	-	-	12	100%
No	18	100%	-	-
Flu vaccine status in 2020				
Vaccinated from flu	13	72%	3	25%
Not vaccinated from flu	5	28%	9	75%

* Refers to having a family member, household member, friend, neighbor, or work colleague with a positive COVID-19 test since January 2020.

**Table 3 healthcare-11-01819-t003:** Key Findings by Theoretical Construct.

Construct	Latino—General	Latino—Migrant Worker
Perceived susceptibility	Low susceptibility due to a high perception of being in good health.	High susceptibility due to the high number of COVID-19 cases and deaths in their community.
Perceived severity	High severity due to the impact on social indicators, such as children’s education, socialization of older adults, and employment.	High severity due to the impact of COVID-19 on health, education, and employment status, and income.
Perceived barriers/self-efficacy	Low barriers to accessing testing sites.	Low barriers to accessing testing sites at workplace.
	High barriers to accessing testing sites for older adults and individuals with disabilities, due to lack of transportation and low proficiency with technology.	High barriers to accessing testing information in Spanish.
	High barriers due to vaccine hesitancy: fear of vaccine side effects, mistrust in the vaccine development process, and uncertainty about its effectiveness.	High barrier due to fear of vaccine’s long-term effects.
	High barriers to accessing the vaccine, due to eligibility criteria at the time of the study.	High barriers for testing and vaccination due to fear regarding their legal status.
Perceived benefits	Protecting family members and stopping the pandemic.	Protecting their family and community.
Cues to action	Triggers to testing and vaccination: Visiting elderly relatives Employment and travel requirements	Triggers to testing and vaccination: Peer pressure from family and co-workers Experiencing symptoms Known exposure

**Table 4 healthcare-11-01819-t004:** Campaign materials produced per audience.

Campaign Materials	General	Migrant
Posters	4 (2 English/2 Spanish)	2 (1 English/1 Spanish)
Live read 30-s radio scripts	2 (1 English/1 Spanish)	2 (Spanish)
30-s videos	1 (1 English/1 Spanish)	-
Social media images	12 (6 English/6 Spanish)	2 (Spanish)
WhatsApp GIFs	-	2 (Spanish)
Repository of creative materials	1	1
Guidance on image and copy pairing	1	1
Guidance on image and copy use and dissemination	1	1

## Data Availability

The data presented in this study are available on request from the corresponding author. The data are not publicly available due to privacy to participants from vulnerable communities.

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
