# Peer review of "A Tale of Two Audiences: Formative Research and Campaign Development for Two Different Latino Audiences, to Improve COVID-19 Prevention Behavior"

_healthcare, 2023, doi:10.3390/healthcare11131819_

Round 1

Reviewer 1 Report

Dear Authors!

Congratulation for this paper, which is a good example of the practical useability of research results.

I have some minor issues to comment on:

1. In the Methodology section, please address the issue that the sample size of the two populations is different.

2. Do not forget to have at least one reference for the Health Belief Model!

3. The Abstract already mentions, that attitudes towards vaccine were different, in the sense that the general Latino population was more reluctant to get vaccinated. However, in the table on Socio-demographic characteristics of the two focus groups, it comes out that more general Latinos were vaccinated in 2020, although it would have been the migrant workers whose jobs required that. Please address this issue.

4. Many quotations are used, which are indeed the strength of the paper and any paper with qualitative methodology. Please identify the respondents with 2 or three features, in brackets, like: "....." (Latino migrant worker, man, aged 36-45). It will confer better readability to the text, and it is also a requirement when using text excerpts. It would be also interesting to find out whether people with lower of higher education had different ideas and thoughts on vaccination, so to differentiate a little bit, or at least to address this issue with some sentences.

5. Finally, some remarks on editing: Please situate Table 3 only after the paragraph on Cues to action.

In the sub-chapter Perceived barriers two of them are used like this: Vaccine hesitancy:...,  Reporting on immigration status:...., while the rest are formulated in sentences. Please rephrase these two also in sentences. In the table on page 6, hyphen ( - ) is missing in the age groups: 18 24 years.

Good luck with the additions.

Best wishes,

reviewer

Author Response

Tale of two audiences: Formative research and campaign development for two different Latino audiences to improve COVID-19 prevention behavior

Response to Reviewer 1 Comments

Point 1: In the Methodology section, please address the issue that the sample size of the two populations is different.

Response 1: We added language addressing this issue on line 186.

Point 2: Include at least one reference for the Health Belief Model.

Response 2: Citation #17, line 576, in the bibliography references the model. Two additional citations references (#18 and #19, lines 578-582) how the model has been used specifically for COVID-19 research.

Point 3: The Abstract already mentions, that attitudes towards vaccine were different, in the sense that the general Latino population was more reluctant to get vaccinated. However, in the table on Socio-demographic characteristics of the two focus groups, it comes out that more general Latinos were vaccinated in 2020, although it would have been the migrant workers whose jobs required that. Please address this issue.

Response 3: This is a good point. Table 2 indicated the percentage of participants that received the flu vaccine. So, although most general Latins received the flu vaccine, they showed hesitancy in getting the COVID-19 vaccines. Whereas most migrant workers did not get the flu vaccine but were more open to the COVID-19 vaccine due to the factors explained in the article. We renamed the categories in Table 2 to clarify the type of vaccine they received.

Point 4: Identify the respondents with 2 or three features, in brackets, like: "....." (Latino migrant worker, man, aged 36-45).

Response 4: Additional demographic descriptors were added throughout. The data has been deidentified and no longer linked to the age of individual participants, so this information is not available.

Point 5: Situate Table 3 only after the paragraph on Cues to action.

Response 5: We moved Table 3 to after ‘Cues to action’ sub-section.

Point 6: In the sub-chapter Perceived barriers two of them are used like this: Vaccine hesitancy:...,  Reporting on immigration status:...., while the rest are formulated in sentences. Please rephrase these two also in sentences. In the table on page 6, hyphen ( - ) is missing in the age groups: 18 24 years.

 Response 6: We edited the three subheadings for consistency. We added the hyphen in all age ranges in Table 2. This may have been removed during the journal’s formatting.

Reviewer 2 Report

A brief summary 

Thank you for giving me the opportunity to review this interesting and important study.

The paper aims to provide empirical evidenced used for developing tailor-made COVID-19 vaccine promotion materials aimed at two different groups of Latino audiences in the US: “general Latinos” and Latinos of the workers´ class. This was done by conducting series of focus groups and qualitatively analysing their contents. The materials were developed and tested in a consecutive phase of the study. Development of tailor-made health promotion materials that take audiences’ background and socio-economic living environment is important. The paper makes contribution to raising awareness on limits to generic health promotion messages that do not take these factors into consideration.   

While the study is highly relevant and important as health promotion materials developed with focus only on individual behaviours have limited effect, it has some major methodological weaknesses.

Although the methodology used is a qualitative study, the results are predominantly reported as in frequency measures (ex. “more often”, “more likely to”). The sample size is too small for making such generalisations and this should not be the aim of a qualitative study. The method used in the study is only appropriate with the prospect of developing a survey questionnaire. I recommend the authors to focus on developing concepts and not counting frequencies to make general statements.

The authors claim that the manuscript covers only Phase 1 (focus groups) but the Phase 2 and 3 are also included in the manuscript, but with very limited information (eye-tracking and facial-expression monitoring and open-ended reflection questions). It is not clear where the current study starts and ends. The aim was to develop the material OR to provide conceptual information that can be tested for phase 2 and 3. This distinction has to be made clear.

Detail comments

Introduction

The references cited in the introduction are recent but many of them are from non-peer reviewed sources. Most of the links provided do not work. It is recommended that the authors replace these references with peer-reviewed articles and/or official government reports and statistics, and check the links. No self-citations are made.

Line 55-57: Citing a 2018 article based on a survey conducted in 2016, the authors conclude: “This was likely due to Latinos’ use of social media, mobile apps, and other digital platforms increasing more than that of the general U.S. population during the COVID-19 pandemic (Flores et al., 2018) [9]”. What the 2016 concludes is that the majority of Hispanic Millennials use internet-based news, and the share is higher than other racial groups because of the higher proportion of Hispanic Millennials in the US in general, compared to other racial groups.

A stronger argument or different statistics are needed to say that the use of social media, etc., increased more than general U.S. population (or other ethnic groups) during the COVID-19 pandemic. Alternatively, the authors can state that the increase in trend of high digital media use could be higher among this specific ethnic group because of their demographic profile (high proportion of Millennials), making this ethnic groups special as compared to others.

Materials and methods

Line 131-133: “To be considered a “migrant worker” specifically, self-identified Latinos also had to work in either farming, agriculture, meatpacking, or the dairy industry”. Please clarify if they have included or excluded Latinos in managerial positions for the second group (migrant workers).

Line 126-127. Please revise the ethics statements with complete information: “Our study protocol was approved by the [NAME OF SCHOOL OF MEDICINE] Institutional Review Board (IRB Protocol 1645842-2)”

All limitations mentioned in the discussion section are critical and are reasons for the methodological weakness of the study. I suggest the authors to mention how they overcome/take into consideration of these problems.  

Please also justify why the proportion of Latino millennia (18-35 years old) is much higher in the Latino general group (33%) than the Latino migrant worker group (8 %). Since the use of social and digital media (especially among the youth) is highlighted in the introduction, I recommend the authors to bring this up either in the result or in the discussion section as a limitation or potential bias.

Results

As mentioned above, the data is predominantly analysed using frequency measures. The aim of the qualitative analysis should be developing categories and also developing relationships between these categories. Some generalisations are not empirically sound. For example, general Latinos could also assume to be suffering from granddaughters not being able to visit them (line 228-231), especially if many older general Latinos were interviewed. It could be a coincident that general Latinos had no first-hand experiences of relatives or people in their social network dying from COVID-19 in this specific sample population. Here, the authors should not treat what the people say as “evidences” or “facts” in a statistical manner but concentrate on their perceptions and what they claim to do or not to do.  

As stated in the limitation the timing of the interviews of the two groups are different. This is quite critical and the authors need to justify how they have accounted for this.

Line 415-416: “However, from the audience profile we know that migrant workers also help support family members on both sides of the border” – This should not be presented as a focus group finding. Please replace it with statements made by the participant.

For the category “communication channels”: the categories seem to saturate but for other categories, it is not clear if and how the saturation has been met.

The issue of immigration status is a different category from general versus workers populations. Documented versus non-document migrants could need different health promotion materials which could be true among other ethnic groups. I suggest the authors to rephrase their findings.

Discussion

As mentioned earlier, I suggest the authors to elaborate on how they intend to overcome the limitations and how their findings are still valid, given all the limitations listed.

There are very little references and literature used here to compare the authors´ findings with previous or similar research. I recommend that they add this too.

Overall, I think the study is very important but there are methodological concerns as listed in this review.

Although English is not my mother tongue, I suggest the author to run another English proofreading. It will improve the quality of the manuscript.

Author Response

Response to Reviewer 2 Comments

Point 1: Although the methodology used is a qualitative study, the results are predominantly reported as in frequency measures (ex. “more often”, “more likely to”). The sample size is too small for making such generalisations and this should not be the aim of a qualitative study. The method used in the study is only appropriate with the prospect of developing a survey questionnaire. I recommend the authors to focus on developing concepts and not counting frequencies to make general statements.

Response 1: We rephrased the three “likely” mentions to reflect the qualitative nature of the study. The edits can be found highlighted on Lines 270, 351, and 360.

Point 2: The authors claim that the manuscript covers only Phase 1 (focus groups) but the Phase 2 and 3 are also included in the manuscript, but with very limited information (eye-tracking and facial-expression monitoring and open-ended reflection questions). It is not clear where the current study starts and ends. The aim was to develop the material OR to provide conceptual information that can be tested for phase 2 and 3. This distinction has to be made clear.

Response 2: We reorganized the text between lines 99-106 and added text to clarify the scope of the article.

Point 3: The references cited in the introduction are recent but many of them are from non-peer reviewed sources. Most of the links provided do not work. It is recommended that the authors replace these references with peer-reviewed articles and/or official government reports and statistics, and check the links. No self-citations are made.

Response 3: Most of the non-peer-reviewed articles included were published by highly reputable government agencies (Centers for Disease Control and Prevention, Food and Drug Administration, and Morbidity and Mortality Weekly Report/CDC). Non-government resources were included if they were published by agencies that provide high quality references publications on specific topics (Pew Research and Migration Policy Institute). Reference #16 is a master’s thesis which we included because we did not find more up to date research publications that spoke to the hyper segmentation of the Latino market.

The following references were replaced:
Reference #4: we included the original research report by Radley, D. et al.

Reference #5: we included an alternative article published in CDC’s MMWR by Dyal, J.W.

Reference #11: we replaced the original reference with a peer-reviewed systematic review article by Rocha, Y.M, et al.

Point 4: Line 55-57: Citing a 2018 article based on a survey conducted in 2016, the authors conclude: “This was likely due to Latinos’ use of social media, mobile apps, and other digital platforms increasing more than that of the general U.S. population during the COVID-19 pandemic (Flores et al., 2018) [9]”. What the 2016 concludes is that the majority of Hispanic Millennials use internet-based news, and the share is higher than other racial groups because of the higher proportion of Hispanic Millennials in the US in general, compared to other racial groups.

A stronger argument or different statistics are needed to say that the use of social media, etc., increased more than general U.S. population (or other ethnic groups) during the COVID-19 pandemic. Alternatively, the authors can state that the increase in trend of high digital media use could be higher among this specific ethnic group because of their demographic profile (high proportion of Millennials), making this ethnic groups special as compared to others.

Response 4: After careful consideration, we removed the mention of social media from the background section. We added a reference to support the statement of information perceptions during COVID-19 which aligns more strongly to the preceding statements. The new reference is #9 by Nagler, et al, 2020.

Point 5: Line 131-133: “To be considered a “migrant worker” specifically, self-identified Latinos also had to work in either farming, agriculture, meatpacking, or the dairy industry”. Please clarify if they have included or excluded Latinos in managerial positions for the second group (migrant workers).

Response 5: An academic project partner was responsible for recruiting migrant workers as they provided direct service to this population. They specifically recruited migrants working in the field and not in managerial positions. We added a clarification in line 135.

Point 6: Line 126-127. Please revise the ethics statements with complete information: “Our study protocol was approved by the [NAME OF SCHOOL OF MEDICINE] Institutional Review Board (IRB Protocol 1645842-2)”

Response 6: We initially left the name of the school blinded for the review process. However, we just added the name of the school in line 121.

Point 7: I suggest the authors mention how they overcome/take into consideration of these problems.  

Response 7: We expanded the discussion and limitations sections. There were some limitation we could not overcome such as the timing of the rollout of the vaccine.

Point 8: Please also justify why the proportion of Latino millennia (18-35 years old) is much higher in the Latino general group (33%) than the Latino migrant worker group (8 %). Since the use of social and digital media (especially among the youth) is highlighted in the introduction, I recommend the authors to bring this up either in the result or in the discussion section as a limitation or potential bias.

Response 8: We removed mentions to social media since it was not the focus of the study.

Point 9: As mentioned above, the data is predominantly analysed using frequency measures. The aim of the qualitative analysis should be developing categories and also developing relationships between these categories. Some generalisations are not empirically sound. For example, general Latinos could also assume to be suffering from granddaughters not being able to visit them (line 228-231), especially if many older general Latinos were interviewed. It could be a coincident that general Latinos had no first-hand experiences of relatives or people in their social network dying from COVID-19 in this specific sample population. Here, the authors should not treat what the people say as “evidences” or “facts” in a statistical manner but concentrate on their perceptions and what they claim to do or not to do.  

Response 9: We have reviewed the results section and changed the language as needed to clarify that these are findings as reported by participants and not facts. We made changes throughout the section as highlighted in yellow.

Point 10: As stated in the limitation the timing of the interviews of the two groups are different. This is quite critical and the authors need to justify how they have accounted for this.

Response 10: We expand on this limitation in the Discussion section.

Point 11: Line 415-416: “However, from the audience profile we know that migrant workers also help support family members on both sides of the border” – This should not be presented as a focus group finding. Please replace it with statements made by the participant.

Response 11: The development of concepts presented in Phase 2 were informed by a combination of data from the focus groups with additional economic and employment data to paint a fuller picture of each audience. The concept “You’re their Rock” is based on the understanding that migrant workers are motivated by supporting their families. However, this is not presented as a direct focus group finding, but as the conclusion after expanding the focus group findings with a complete audience profile. We have added some clarification to this on lines 376-381, and in the lines 507-517.

Point 12: For the category “communication channels”: the categories seem to saturate but for other categories, it is not clear if and how the saturation has been met.

Response 12: Participants mentioned multiple communication preferences and channels throughout the discussions. We summarized those reported more often and consistently. A few edits were made to that section clarifying this point.

Point 13: The issue of immigration status is a different category from general versus workers populations. Documented versus non-document migrants could need different health promotion materials which could be true among other ethnic groups. I suggest the authors rephrase their findings.

Response 13: Immigration status was an issue uniquely cited by migrant workers as they perceived it as a barrier to accessing clinics to get tested or to get a vaccine. These findings were not related to the type or format of health promotion materials they would receive. Please clarify if there is a different section of the manuscript we need to discuss the immigration status.

Point 14: As mentioned earlier, I suggest the authors elaborate on how they intend to overcome the limitations and how their findings are still valid, given all the limitations listed.

Response 14: We expanded the discussion and limitations sections to describe how we overcame them.

Point 15: There are very little references and literature used here to compare the authors´ findings with previous or similar research. I recommend that they add this too.

Response 15: We added additional reflections on how the findings compare to similar research. We added new references (20,21, and 22).

Round 2

Reviewer 2 Report

Dear authors,

I think the quality of the manuscript has improved. Congratulations to your work.